# Blood Pressure Levels and Triglyceride–Glucose Index: A Cross-Sectional Study from a Nationwide Screening in Mongolia

**DOI:** 10.3390/jcm14196890

**Published:** 2025-09-29

**Authors:** Byambasuren Dagvajantsan, Oyunsuren Enebish, Khangai Enkhtugs, Bayarbold Dangaa, Munkhtulga Gantulga, Mijidsuren Ganbat, Narantuya Davaakhuu, Tumur-Ochir Tsedev-Ochir, Batzorig Bayartsogt, Enkhtur Yadamsuren, Altantuya Shirchinjav, Oyuntugs Byambasukh

**Affiliations:** 1Department of Neurology, School of Medicine, Mongolian National University of Medical Sciences, Ulaanbaatar 14210, Mongolia; byambasuren@mnums.edu.mn; 2Ministry of Health, Ulaanbaatar 14253, Mongolia; oyunsuren@moh.gov.mn (O.E.); bayarbold@moh.gov.mn (B.D.); 3Department of Family Medicine, School of Medicine, Mongolian National University of Medical Sciences, Ulaanbaatar 14210, Mongolia; khangai@mnums.edu.mn; 4Department of Epidemiology and Biostatistics, School of Public Health, Mongolian National University of Medical Sciences, Ulaanbaatar 14210, Mongolia; batzorig@mnums.edu.mn; 5State Central Third Hospital, Ulaanbaatar 16081, Mongolia; munkhtulga@msic.mn (M.G.); contact@msic.mn (M.G.); cardio@telemedicine.mn (N.D.); tsch@shastinhospital.mn (T.-O.T.-O.); 6Deprtment of Dermatology, School of Medicine, Mongolian National University of Medical Sciences, Ulaanbaatar 14210, Mongolia; enkhtur@mnums.edu.mn; 7Department of Endocrinology, School of Medicine, Mongolian National University of Medical Sciences, Ulaanbaatar 14210, Mongolia

**Keywords:** insulin resistance, hypertension, central obesity, Mongolia

## Abstract

**Background**: The triglyceride–glucose (TyG) index has emerged as a reliable surrogate marker of insulin resistance. This study aimed to investigate the association between blood pressure (BP) levels and the TyG index and to assess whether even modest elevations in BP were associated with higher TyG index values. **Methods**: A cross-sectional analysis was conducted using data from 120,264 participants who underwent nationwide health screening in Mongolia between 2023 and 2024. BP was categorized into five stages. The TyG index was calculated based on fasting triglyceride and glucose levels. **Results**: The mean TyG index increased progressively with advancing hypertension stages (*p* for trend <0.001). Multivariate analysis showed that even elevated BP was independently associated with a higher TyG index (adjusted OR 1.108, 95% CI 1.039–1.183; *p* = 0.002), with the association strengthening across hypertension stage 1 (adjusted OR 1.238, 95% CI 1.200–1.277), stage 2 (adjusted OR 1.516, 95% CI 1.463–1.572), and hypertensive crisis (adjusted OR 1.575, 95% CI 1.350–1.836) (all *p* < 0.001). Central obesity further amplified the association between hypertension stage and TyG index levels. Among participants without central obesity, the TyG index increased from 8.086 (95% CI: 8.079–8.093) in the normal BP group to 8.449 (8.362–8.536) in the hypertensive crisis group. Similarly, among those with central obesity, the TyG index rose from 8.345 (8.336–8.354) in the normal group to 8.732 (8.685–8.778) in the hypertensive crisis group. **Conclusions**: This study demonstrates that the TyG index rises consistently with increasing BP stages, even at early elevations, suggesting that insulin resistance may begin at modest blood pressure increases.

## 1. Introduction

Hypertension and insulin resistance are widely recognized as interrelated components of metabolic dysfunction, both of which contribute substantially to the global burden of non-communicable diseases [1,2]. Together, these conditions increase the risk of cardiovascular disease, diabetes, and non-alcoholic fatty liver disease (NAFLD), conditions that are becoming increasingly prevalent in both developed and developing countries [3,4]. Hypertension has traditionally been regarded as a late manifestation of vascular and metabolic dysfunction, often emerging after prolonged exposure to adverse lifestyle and metabolic risk factors. However, accumulating evidence indicates that metabolic abnormalities, particularly insulin resistance, may precede or accompany even modest increases in blood pressure, suggesting that vascular changes may begin much earlier in the disease process [1,2,5,6]. This highlights the importance of identifying simple, reliable markers that can capture early metabolic derangements before the onset of overt hypertension and its complications. Mechanistically, insulin resistance may contribute to blood pressure elevation through sympathetic nervous system activation, enhanced renal sodium retention, endothelial dysfunction, and impaired glucose uptake in peripheral tissues, while elevated blood pressure itself can exacerbate insulin resistance, supporting a bidirectional relationship [3,7].

The triglyceride–glucose (TyG) index, calculated from routine fasting triglyceride and glucose levels, has emerged as a practical and reproducible surrogate marker for insulin resistance [8]. Compared with the hyperinsulinemic–euglycemic clamp—the gold standard but technically demanding method for assessing insulin sensitivity—the TyG index offers a simpler, cost-effective, and widely applicable alternative. Its ease of calculation makes it especially valuable in large-scale epidemiological and population-based studies, where invasive and resource-intensive methods are not feasible [8,9,10]. Previous studies have consistently shown that elevated TyG index values are associated with metabolic syndrome, future cardiovascular events, and the development of hypertension [11,12,13]. Beyond metabolic influences, recent evidence suggests that the TyG index may also capture genetic predisposition to insulin resistance, with higher genetic risk linked to reduced longevity and increased cardiometabolic risk. Incorporating both metabolic and genetic dimensions enhances the relevance of the TyG index as a marker of early cardiometabolic dysfunction [14,15]. Despite this growing body of evidence, the relationship between graded increases in blood pressure—from normal to elevated, and then to hypertensive stages—and TyG index levels has not been fully elucidated. Moreover, this relationship has been insufficiently characterized across different ethnic, geographic, and socioeconomic populations, many of which are undergoing rapid health transitions.

Mongolia represents one such setting. The country is currently experiencing a rapid epidemiological shift marked by rising rates of hypertension, diabetes, obesity, and cardiovascular disease, driven by urbanization, dietary changes, and sedentary lifestyles [16,17,18]. In this context, national screening programs provide a valuable opportunity to investigate metabolic risk markers at the population level. The 2023–2024 nationwide health screening in Mongolia, one of the largest and most representative to date, enables the exploration of how common metabolic indices such as the TyG index relate to early cardiovascular risk indicators in this unique population. Understanding whether even modest elevations in blood pressure are associated with evidence of insulin resistance could provide important insights for risk stratification, early prevention strategies, and tailored interventions in countries facing high and rising burdens of cardiometabolic disease.

Therefore, the present study aimed to examine the association between blood pressure categories and the TyG index in a nationally representative sample of Mongolian adults. We specifically hypothesized that even early elevations in blood pressure—prior to the onset of clinical hypertension—would be associated with higher TyG index values, reflecting the early emergence of insulin resistance in this high-risk population.

## 2. Materials and Methods

### 2.1. Study Participants

This cross-sectional study was based on data from the Mongolian nationwide health screening program, which was conducted between 2023 and 2024. The design and implementation of the program, including detailed procedures for recruitment and data collection, have been described previously [19]. Briefly, the program targeted adults across all regions of Mongolia, providing standardized health examinations and laboratory testing.

All individuals aged 18 years and older who participated in the screening were initially eligible for inclusion. Exclusion criteria were applied to ensure data reliability and to minimize confounding. Specifically, participants were excluded if they had (i) missing or extreme outlier values for fasting plasma glucose, triglycerides, or anthropometric measurements; (ii) a self-reported history of cardiovascular disease or chronic kidney disease; or (iii) current pregnancy at the time of screening. After applying these criteria, a total of 120,264 participants were included in the final analysis.

The study protocol adhered to the principles of the Declaration of Helsinki and was approved by the Medical Ethics Committee of the Ministry of Health, Mongolia (Approval No: 23/042, dated 5 July 2023).

### 2.2. Data Collection and Study Variables

Data were obtained from the national screening database, which included both self-reported information and clinically measured variables [19]. Demographic variables included age, sex, education level, smoking status, alcohol use, and physical activity. Clinical data collected during the examination comprised anthropometric measures, blood pressure, and laboratory tests.

Anthropometric assessment: Height and weight were measured using calibrated instruments, and body mass index (BMI) was calculated as weight (kg) divided by height (m^2^). BMI was classified as normal weight (18.5–24.9 kg/m^2^), overweight (25.0–29.9 kg/m^2^), or obese (≥30.0 kg/m^2^). Waist circumference was measured at the midpoint between the lower rib and the iliac crest. Central obesity was defined as ≥90 cm in men or ≥80 cm in women, according to international criteria.

Blood pressure measurement: Blood pressure was measured twice after the participant had been seated at rest for at least five minutes using calibrated automated sphygmomanometers (Omron Healthcare, Kyoto, Japan), and examiners received standardized training. To minimize inter-observer variability, repeated measurements were averaged, consistent with the national screening protocol. Hypertension was defined as systolic blood pressure (SBP) ≥140 mmHg, diastolic blood pressure (DBP) ≥90 mmHg, or current use of antihypertensive medication. Participants were further classified into five categories according to American Heart Association (AHA) guidelines [20]:Normal (SBP < 120 mmHg and DBP < 80 mmHg)Elevated (SBP 120–129 mmHg and DBP < 80 mmHg)Hypertension stage 1 (SBP 130–139 mmHg or DBP 80–89 mmHg)Hypertension stage 2 (SBP ≥ 140 mmHg or DBP ≥ 90 mmHg)Hypertensive crisis (SBP ≥ 180 mmHg or DBP ≥ 120 mmHg)

Laboratory measurements and TyG index: Fasting plasma glucose and triglyceride concentrations were measured using standardized enzymatic methods (Hitachi 7600 Analyzer, Hitachi High-Technologies, Tokyo, Japan). The triglyceride–glucose (TyG) index was calculated as [21]:$$\mathrm{TyG}\;\mathrm{index}\;=\ln\left(\frac{\;triglycerides\;(\mathrm{mg}/\mathrm{dL})\times glucose\;(\mathrm{mg}/\mathrm{dL})}{2}\right)$$

Original values (in mmol/L) were converted to mg/dL using standard conversion factors (triglycerides ×88.5; glucose ×18). For analysis, the TyG index was dichotomized into two groups: a lower TyG group (below the median) and a higher TyG group (at or above the median).

Diabetes status: Diabetes was defined as fasting plasma glucose ≥7.0 mmol/L or current use of glucose-lowering medication.

### 2.3. Statistical Analysis

Continuous variables are summarized as mean ± standard deviation (SD), while categorical variables are presented as frequencies and percentages. Comparisons between hypertensive and normotensive groups were performed using independent-samples *t*-tests for continuous variables and chi-square tests for categorical variables.

To examine differences in the TyG index across the five AHA-defined blood pressure categories, analysis of variance (ANOVA) was applied, followed by Bonferroni correction for multiple comparisons.

The correlation between SBP and the TyG index was assessed using Pearson correlation coefficients. The association between blood pressure levels and high TyG index (binary classification) was assessed using logistic regression models. Both univariate and multivariate models were performed. The multivariate models included adjustments for key covariates: age, sex, waist circumference, and diabetes status. A stepwise modeling approach was used: Model 1 was unadjusted; Model 2 was adjusted for age and sex; Model 3 was additionally adjusted for waist circumference and diabetes status; and Model 4 was further adjusted for education, living area and smoking. In addition to categorical analyses based on AHA-defined blood pressure groups, logistic regression models were also fitted using systolic blood pressure (SBP) as a continuous variable to assess its association with increased TyG index. Results are expressed as odds ratios (ORs) with 95% confidence intervals (CIs). To evaluate the discriminative performance of systolic blood pressure (SBP) in identifying individuals with elevated TyG index, we performed receiver operating characteristic (ROC) curve analysis.

As a sensitivity analysis, we further examined the relationship between SBP and the TyG index by treating both variables as continuous. Linear regression models were stratified by blood pressure status (normotensive vs. hypertensive) to assess whether the strength of association differed across groups.

All statistical analyses were carried out using IBM SPSS Statistics, Version 28.0 (IBM Corp., Armonk, NY, USA). A *p*-value < 0.05 was considered statistically significant.

## 3. Results

A total of 120,264 participants were included in the final analysis. The mean age of the population was 45.1 ± 15.2 years, and 39.4% were male. When stratified by blood pressure category, 35.3% were classified as normotensive, 4.0% as having elevated blood pressure, 32.5% as hypertension stage 1, 27.4% as hypertension stage 2, and 0.7% as hypertensive crisis. Thus, nearly two-thirds of participants fell into a hypertensive category.

Table 1 summarizes the clinical and biochemical characteristics of the study population by hypertension status. Compared with normotensive individuals, those with hypertension were significantly older (48.2 ± 14.6 vs. 37.0 ± 13.5 years, *p* < 0.001) and more likely to be male (42.2% vs. 34.2%, *p* < 0.001). Lifestyle factors also differed: the prevalence of current smoking was higher among hypertensive participants (20.5% vs. 17.8%, *p* < 0.001).

Marked differences in anthropometric measures were observed. Hypertensive participants exhibited higher mean BMI (27.6 ± 4.9 vs. 24.9 ± 4.3 kg/m^2^, *p* < 0.001) and waist circumference (89.2 ± 14.1 vs. 81.6 ± 12.6 cm, *p* < 0.001). Both central obesity (63.5% vs. 41.9%, *p* < 0.001) and general obesity (28.7% vs. 11.7%, *p* < 0.001) were substantially more prevalent in the hypertensive group. These findings highlight the close relationship between excess adiposity and elevated blood pressure.

Biochemical differences paralleled the clinical characteristics. Hypertensive individuals demonstrated significantly higher fasting plasma glucose (5.24 ± 1.54 vs. 4.85 ± 0.99 mmol/L, *p* < 0.001) and triglyceride levels (1.38 ± 0.94 vs. 1.12 ± 0.79 mmol/L, *p* < 0.001). Consistent with these findings, the mean TyG index was also elevated in the hypertensive group (8.47 ± 0.64 vs. 8.19 ± 0.59, *p* < 0.001). Furthermore, the prevalence of diabetes was more than twice as high among participants with hypertension compared with those with normal blood pressure (9.0% vs. 3.6%, *p* < 0.001).

Pearson correlation analysis showed a modest but statistically significant positive association between systolic blood pressure and the TyG index (r = 0.256, *p* < 0.001; Figure 1). A stepwise increase in TyG index was observed across blood pressure categories (*p* for trend <0.001; Figure 1). The mean TyG index rose from 8.195 (95% CI, 8.189–8.201) in the normotensive group to 8.302 (95% CI, 8.285–8.320) in the elevated BP group, 8.382 (95% CI, 8.376–8.388) in stage 1 hypertension, 8.597 (95% CI, 8.591–8.604) in stage 2 hypertension, and 8.673 (95% CI, 8.632–8.715) among participants in hypertensive crisis.

Univariate logistic regression revealed progressively higher odds of elevated TyG index across increasing blood pressure stages, with normotension as the reference category (Table 2). The unadjusted odds ratios (ORs) were 1.412 (95% CI, 1.328–1.501) for elevated BP, 1.693 (95% CI, 1.646–1.742) for stage 1, 3.055 (95% CI, 2.965–3.148) for stage 2, and 3.806 (95% CI, 3.303–4.386) for hypertensive crisis. After stepwise adjustment for potential confounders, including age, gender, waist circumference, and diabetes, the associations remained robust. In the fully adjusted model (age, gender, waist circumference, and diabetes, education, living area and smoking), elevated BP (OR 1.108, 95% CI 1.039–1.183), stage 1 hypertension (OR 1.238, 95% CI 1.200–1.277), stage 2 hypertension (OR 1.516, 95% CI 1.463–1.572), and hypertensive crisis (OR 1.575, 95% CI 1.350–1.836) were each independently associated with a higher TyG index.

When analyzed as a continuous variable, systolic blood pressure was consistently associated with higher odds of elevated TyG index across all models. In the fully adjusted model, each 1 mmHg increase in SBP was associated with a 1.0% higher odds of elevated TyG index (OR 1.010, 95% CI 1.009–1.010, *p* < 0.0001; Table 2). These findings were consistent with the stepwise associations observed in categorical BP analyses. Furthermore, ROC curve analysis showed that SBP had modest discriminative ability for elevated TyG index, with an AUC of 0.641 (Appendix A).

In sensitivity analyses using SBP and TyG as continuous variables, a positive linear association was observed in both normotensive and hypertensive participants. Among normotensives, each 1 mmHg increase in SBP was associated with a 0.003-unit higher TyG index. The association was slightly stronger in hypertensives, with each 1 mmHg increase in SBP corresponding to a 0.005-unit higher TyG index (Table 3).

An interaction analysis demonstrated a significant modifying effect of central obesity on the association between blood pressure stage and TyG index (overall interaction *p* = 0.015), particularly at stage 2 (interaction *p* = 0.006). Subgroup analyses stratified by central obesity status (Figure 2) showed that although the TyG index increased progressively across blood pressure stages in both groups, individuals with central obesity consistently had higher absolute values. Among those without central obesity, the mean TyG index rose from 8.086 in the normotensive group to 8.449 in hypertensive crisis. In comparison, among those with central obesity, values ranged from 8.345 in the normotensive group to 8.732 in hypertensive crisis.

Together, these findings indicate that increasing blood pressure stages are strongly and independently associated with a higher TyG index, and that central obesity accentuates the absolute burden of insulin resistance across the hypertension spectrum.

## 4. Discussion

In this nationwide cross-sectional study, we demonstrated a progressive relationship between blood pressure levels and the TyG index, a surrogate marker of insulin resistance, in a large Mongolian population. Notably, even individuals categorized as having elevated blood pressure, a stage below clinical hypertension, exhibited a significantly higher TyG index compared with normotensive participants. These findings highlight that metabolic disturbances likely begin early in the blood pressure continuum and support the hypothesis that elevated blood pressure and insulin resistance are closely intertwined well before the onset of overt hypertension. Our results showed that hypertensive participants were older, more frequently male, and exhibited higher BMI, waist circumference, fasting glucose, triglycerides, and TyG index compared with normotensive individuals. A progressive increase in TyG index was observed across ascending blood pressure stages, ranging from 8.195 among normotensive individuals to 8.673 among those experiencing hypertensive crisis. The association remained significant even after adjusting for potential confounders, including age, sex, waist circumference, and diabetes, suggesting an independent link between blood pressure elevation and underlying insulin resistance. Importantly, even participants with elevated blood pressure (120–129/<80 mmHg) had a statistically significant increase in TyG index, with 10.8% higher odds of elevated TyG compared with those with normal blood pressure. This reinforces the concept that metabolic derangements begin much earlier than the conventional blood pressure thresholds used in clinical practice.

Based on our results, we observed a progressive relationship between increasing blood pressure levels and the TyG index, a surrogate marker of insulin resistance, even among individuals with elevated blood pressure below the threshold for clinical hypertension. These findings are consistent with a large body of evidence demonstrating a strong association between insulin resistance and hypertension. Meta-analyses and observational studies have reported that individuals with higher fasting insulin levels or greater insulin resistance, as measured by the HOMA-IR index, are at substantially increased risk of developing hypertension, with risk ratios ranging from 1.43 to 1.54 for the highest versus lowest categories of insulin resistance [1,2].

This dose-dependent relationship between blood pressure and TyG index has also been consistently demonstrated across populations. In China, individuals with higher blood pressure showed progressively higher TyG values, with those in the top TyG categories experiencing nearly double the risk of developing hypertension [22,23]. Korean studies further support this trend, reporting graded increases in TyG across rising blood pressure levels in both adults and youth [24]. In Mexico, long-term follow-up confirmed that adults with higher blood pressure were more likely to fall into higher TyG categories, and those in the top tertile had a 56% increased risk of hypertension compared to the lowest group [25,26]. Evidence from Saudi Arabia likewise found that individuals with elevated systolic blood pressure had significantly higher TyG values, particularly women and those with obesity [27]. Large-scale data from the UK Biobank extend these findings to European populations, showing a steady rise in cardiovascular risk across TyG quartiles, with blood pressure mediating part of this association [28]. Collectively, these studies reinforce that rising blood pressure is accompanied by dose-dependent elevations in the TyG index, underscoring the robustness and generalizability of this relationship. The relatively high absolute TyG values observed in our Mongolian sample appear higher than those reported in studies above mentioned [22,23,24,25,26,27,28]. This discrepancy may reflect Mongolia’s distinctive dietary and sociocultural patterns, including heavy reliance on red meat and animal fat, limited fruit and vegetable intake, and lifestyle transitions related to rapid urbanization, which together may contribute to higher baseline insulin resistance and consequently elevated TyG levels [18].

Mechanistically, our findings can be explained by several established pathways through which insulin resistance contributes to blood pressure elevation, including activation of the sympathetic nervous system, enhanced renal sodium reabsorption, impaired nitric oxide-mediated vasodilation, and vascular remodeling [3,6,28,29,30]. Supporting this interpretation, a study in Korean adults also demonstrated that higher TyG index values were associated with both prehypertension and hypertension, reinforcing the idea that insulin resistance may precede and promote blood pressure elevation [31]. At the same time, hypertension itself may exacerbate insulin resistance by impairing insulin and glucose delivery to skeletal muscle, a consequence of vascular changes and increased vasoconstriction [28,30]. Moreover, both conditions may share common underlying mechanisms, including chronic inflammation, oxidative stress, and activation of the renin–angiotensin–aldosterone system [3,4,6,29]. Taken together with our results, these findings suggest that the relationship between insulin resistance and blood pressure is bidirectional and already evident at the earliest stages of blood pressure elevation.

In this study, we found that central obesity significantly modified the association between blood pressure stage and the TyG index (overall interaction *p* = 0.015), with the effect most pronounced at stage 2 hypertension (*p* = 0.006). Subgroup analyses demonstrated that although the TyG index increased progressively across blood pressure stages in both centrally obese and non-obese participants, individuals with central obesity consistently had higher absolute values. Among those without central obesity, the mean TyG index rose from 8.086 in normotensive participants to 8.449 in hypertensive crisis, whereas in those with central obesity, the values ranged from 8.345 to 8.732. These results align with previous evidence showing that the TyG index, a well-established surrogate marker of insulin resistance, is positively associated with both blood pressure and indicators of central obesity such as waist circumference and BMI [27,28,29,30]. Central obesity is widely recognized as a powerful amplifier of metabolic risk, as individuals with higher waist circumference or BMI typically exhibit elevated TyG values and a higher likelihood of developing hypertension and cardiovascular disease, irrespective of their blood pressure status [32,33,34,35]. Our findings add to this evidence by showing that the relationship between blood pressure stage and TyG index persists even after accounting for central obesity. Taken together, these findings suggest that both elevated blood pressure and central obesity should be regarded as early indicators of underlying metabolic dysfunction, and that their coexistence likely confers a substantially higher risk of adverse cardiometabolic outcomes.

The strengths of this study include its large and nationally representative sample, the use of standardized measurement protocols, and comprehensive adjustment for key confounders. By analyzing blood pressure stages in detail and their independent associations with the TyG index, this study provides new insights into the metabolic shifts that accompany early blood pressure changes. However, several limitations should be considered. First, the cross-sectional design precludes causal inference, and longitudinal studies are needed to clarify the temporal sequence between insulin resistance and blood pressure elevation. The observed associations may be bidirectional: insulin resistance can contribute to higher blood pressure through mechanisms such as sympathetic activation, sodium retention, and endothelial dysfunction, whereas elevated blood pressure may, in turn, impair insulin-mediated glucose delivery and exacerbate insulin resistance. Thus, the interplay between blood pressure and insulin sensitivity is likely complex and reciprocal. Second, although the TyG index is a validated surrogate marker, it does not directly assess glucose–insulin dynamics and may be influenced by external factors such as alcohol intake or acute illness, which could not be fully controlled. Third, lifestyle factors such as detailed physical activity levels and dietary habits were not extensively evaluated, leaving the possibility of residual confounding. In addition, several potentially relevant covariates, including lipid parameters (HDL-C, LDL-C) and family history of hypertension or diabetes, were not available in the dataset and therefore could not be included in the regression models. This absence may have contributed to residual confounding. Furthermore, medication uses may have influenced both blood pressure and the TyG index, potentially introducing bias. However, information on glucose-lowering and lipid-lowering medication was not systematically collected in the dataset and therefore could not be incorporated into the regression models. This limitation should be considered when interpreting the findings. Fourth, alcohol use was not an exclusion criterion, and while prevalence was relatively modest, it may still have affected triglyceride variability. Fifth, the unique sociocultural and dietary characteristics of the Mongolian population, including high red meat intake and low fruit and vegetable consumption, may limit the generalizability of these findings to other populations. Finally, the hypertension prevalence observed in this study (65%) may be overestimated because blood pressure was assessed using only two readings at a single visit. In contrast, international protocols recommend repeated measurements across multiple visits to confirm the diagnosis. This methodological difference may partly account for the higher prevalence.

Taken together, the progressive rise in TyG index even among individuals with elevated blood pressure suggests that metabolic screening may need to begin earlier than currently practiced. Incorporating the TyG index into routine assessments could provide a simple and cost-effective strategy for early detection of insulin resistance in both primary care and population-level screening, particularly in low-resource settings. From a clinical perspective, even modest increases in the TyG index may be meaningful. Prior studies show that a 0.3–0.5 unit rise is linked to higher risks of hypertension, poor BP control, and cardiovascular events. Thus, the differences observed across BP categories in our study are likely clinically relevant for early risk stratification and prevention. Furthermore, the combined burden of elevated blood pressure and central obesity highlights the importance of integrated interventions targeting both weight control and blood pressure management to reduce future risks of diabetes, cardiovascular disease, and related complications. From a public health perspective, these findings support the incorporation of the TyG index into national screening programs in Mongolia. Early identification of individuals with elevated TyG values, even at modest blood pressure elevations, could help target preventive interventions more effectively. Integrating TyG into routine risk assessment may thus strengthen Mongolia’s capacity to mitigate the growing burden of hypertension and cardiometabolic disease. Public health strategies aimed at promoting healthier dietary patterns, physical activity, and weight reduction may therefore have substantial benefits in preventing both hypertension and metabolic disorders.

## 5. Conclusions

This large, nationally representative study of Mongolian adults demonstrates a strong, progressive association between blood pressure levels and the TyG index, beginning even at the elevated blood pressure stage. Importantly, central obesity was found to significantly modify this relationship, with obese individuals consistently exhibiting higher absolute TyG values across all blood pressure categories, despite a modest attenuation of the incremental effect of rising blood pressure. These findings highlight that insulin resistance and metabolic dysfunction emerge early in the blood pressure continuum and are amplified by abdominal adiposity. Clinically, our results underscore the value of incorporating the TyG index into routine screening, particularly in individuals with elevated blood pressure or central obesity, to enable earlier identification of those at high cardiometabolic risk. Public health strategies that simultaneously address blood pressure control, obesity prevention, and insulin resistance may be critical to reducing the growing burden of hypertension, diabetes, and cardiovascular disease in Mongolia and other high-risk populations.

## Figures and Tables

**Figure 1 jcm-14-06890-f001:**
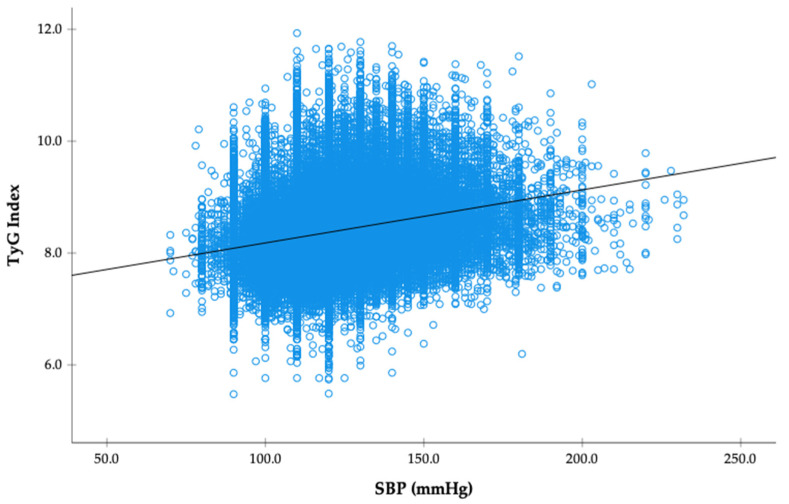
Scatterplot showing the association between SBP and the TyG index in the study population. Each blue circle represents an individual participant, and the black line represents the fitted regression line.

**Figure 2 jcm-14-06890-f002:**
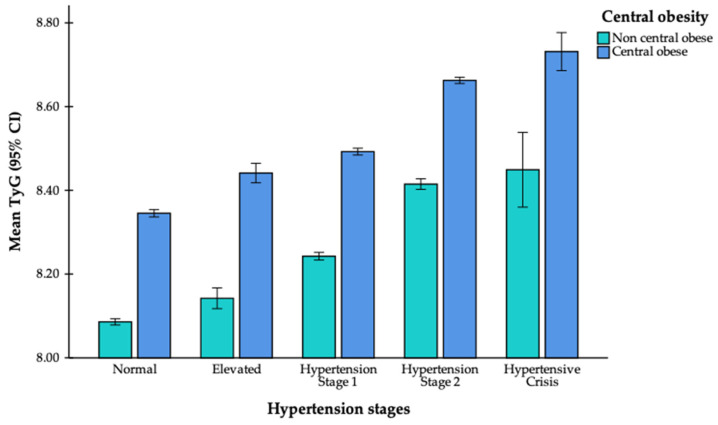
Association between hypertension stage and TyG index by central obesity status. Error bars indicate 95% confidence intervals.

**Table 1 jcm-14-06890-t001:** Characteristics of study population.

Findings	Total (120,264)	Normotensive (42,475)	Hypertensive (77,789)	*p*-Value
Age (years)	45.1 ± 15.2	37.0 ± 13.5	48.2 ± 14.6	<0.001
Male gender, % (n)	39.4 (n = 47,370)	34.2 (n = 14,519)	42.2 (n = 32,851)	<0.001
Current smoker, % (n)	19.5 (n = 23,494)	17.8 (n = 7574)	20.5 (n = 15,920)	<0.001
BMI (kg/m^2^)	26.7 ± 4.8	24.9 ± 4.3	27.6 ± 4.9	<0.001
Waist circumference (cm)	86.6 ± 13.9	81.6 ± 12.6	89.2 ± 14.1	<0.001
Fasting glucose (mmol/L)	5.1 ± 1.4	4.85 ± 0.99	5.24 ± 1.54	<0.001
Triglycerides (mmol/L)	1.3 ± 0.9	1.12 ± 0.79	1.38 ± 0.94	<0.001
TyG index	8.38 ± 0.62	8.19 ± 0.59	8.47 ± 0.64	<0.001
Central obesity, % (n)	55.9 (n = 67,177)	41.9 (n = 17,795)	63.5 (n = 49,382)	<0.001
Obesity, % (n)	22.7 (n = 27,317)	11.7 (n = 4976)	28.7 (n = 22,341)	<0.001
Diabetes, % (n)	7.1 (n = 8515)	3.6 (n = 1537)	9.0 (n = 6978)	<0.001

Data are presented as mean ± SD and percentages (numbers). The *p*-values correspond to ANOVA for continuous variables and Pearson’s chi-square test for categorical variables.

**Table 2 jcm-14-06890-t002:** Regression analysis for the association between BP levels and Increased TyG Index.

	Odds Ratio	95% CI	*p*-Value
Model 1. Unadjusted			
SBP (cont.)	1.029	1.028–1.030	<0.0001
Normotensive	0 (Ref)	-	-
Elevated BP	1.412	1.328–1.501	<0.0001
Hypertension Stage 1	1.693	1.646–1.742	<0.0001
Hypertension Stage 2	3.055	2.965–3.148	<0.0001
Hypertensive Crisis	3.806	3.303–4.386	<0.0001
Model 2. Adjusted for age and gender			
SBP (cont.)	1.017	1.016–1.018	<0.0001
Normotensive	0 (Ref)	-	-
Elevated BP	1.291	1.230–1.355	<0.0001
Hypertension Stage 1	1.619	1.553–1.686	<0.0001
Hypertension Stage 2	2.119	2.029–2.211	<0.0001
Hypertensive Crisis	2.102	1.653–2.671	<0.0001
Model 3. Adjusted for age, gender, waist circumference and DM status			
SBP (cont.)	1.009	1.009–1.010	<0.0001
Normotensive	0 (Ref)	-	-
Elevated BP	1.191	1.123–1.263	<0.0001
Hypertension Stage 1	1.579	1.510–1.651	<0.0001
Hypertension Stage 2	1.655	1.582–1.731	<0.0001
Hypertensive Crisis	1.979	1.601–2.447	<0.0001
Model 4. Adjusted for age, gender, waist circumference, DM status, education, living area and smoking			
SBP (cont.)	1.010	1.009–1.010	<0.0001
Normotensive	0 (Ref)	-	-
Elevated BP	1.108	1.039–1.183	<0.0001
Hypertension Stage 1	1.238	1.200–1.277	<0.0001
Hypertension Stage 2	1.516	1.463–1.572	<0.0001
Hypertensive Crisis	1.575	1.350–1.836	<0.0001

Logistic regression analyses. BP, blood pressure; CI, confidence interval.

**Table 3 jcm-14-06890-t003:** Linear regression analysis of the association between systolic blood pressure (SBP, continuous) and TyG index (continuous) stratified by normotensive and hypertensive groups.

	Odds Ratio	95% CI	*p*-Value
Normotensive			
SBP (cont.)	0.003	0.002–0.004	<0.0001
Hypertensive			
SBP (cont.)	0.005	0.003–0.006	<0.0001

Linear regression analyses. CI, confidence interval.

## Data Availability

The data used to support the findings of this study are available from the corresponding author upon request.

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
