# Peer review of "Blood Pressure Levels and Triglyceride–Glucose Index: A Cross-Sectional Study from a Nationwide Screening in Mongolia"

_jcm, 2025, doi:10.3390/jcm14196890_

Round 1

Reviewer 1 Report

Comments and Suggestions for Authors

This is a well-designed population-based study with valuable clinical and public health implications. The results are robust and the manuscript is structured clearly. However, several important issues must be addressed to strengthen its scientific rigor and readability.

The cross-sectional nature should be more strongly emphasized as it precludes causal inference. Please discuss the potential reverse causality between blood pressure and insulin resistance in greater depth.

Clarify whether BP was measured using automated or manual devices, and if inter-observer variability was controlled. This is critical in a nationwide survey.

The reported prevalence 65% is strikingly high. Discuss whether the methodology (two BP readings at one visit) may have overestimated hypertension prevalence compared with standard epidemiological surveys.

Discussion

Expand comparisons with studies from other Asian and Eastern European countries, highlighting whether the magnitude of association is consistent with other populations.

Provide a more detailed discussion of dietary and sociocultural factors in Mongolia that may explain the relatively high absolute TyG values.

Strengthen the public health perspective: how might these findings influence Mongolian health policies or screening programs?

Figures (especially Figure 1 and 2) should be redesigned for better readability and visual impact.

References:

Update with more recent literature (2022–2024).

Eliminate duplicated citations (e.g., references 8 and 10).

Author Response

We sincerely appreciate the reviewer’s thoughtful comments and constructive suggestions, which have substantially strengthened and refined our manuscript. All revisions are highlighted in yellow in the revised version.

Reviewer 2 Report

Comments and Suggestions for Authors

The manuscript investigates the association between blood pressure levels and the triglyceride-glucose (TyG) index in a large, nationally representative Mongolian cohort. The topic is timely and relevant, particularly given the increasing interest in surrogate markers of insulin resistance. While the study offers valuable insights, several key issues should be addressed to enhance its clarity, methodological rigor, and interpretability.

  1. Scientific Background and Hypothesis
  • The introduction would benefit from a clearer articulation of the hypothesized physiological mechanism linking blood pressure and the TyG index. At present, the causal pathway is only implied.
  • The TyG index is a validated surrogate marker of insulin resistance, influenced not only by metabolic factors but also by genetic predisposition. Recent studies have shown that genetic risk for insulin resistance is associated with reduced longevity and increased cardiometabolic risk. These aspects could be briefly discussed to enrich the manuscript’s implications.
  1. Study Design and Confounding
  • The cross-sectional design inherently limits causal inference. This limitation should be explicitly acknowledged in the “Limitations” section.
  • It is unclear which confounders were included in the multivariate models. Important variables such as physical activity, smoking status, alcohol consumption, dietary habits, and family history of hypertension or diabetes should be considered.
  • Several indicators listed in Table 1 (e.g., BMI, total cholesterol, HDL, LDL, education level, urban/rural residence) are relevant covariates and should be incorporated into the analysis to improve model accuracy and generalizability.
  1. Medication Use and Bias
  • Although diabetes identification included medication use, it is not clear whether glucose-lowering, lipid-lowering, or antihypertensive medications were adjusted for in the regression models.
  • These medications directly influence the components of the TyG index and blood pressure, potentially biasing the observed associations.
  • If data are available, a sensitivity analysis comparing treated vs. untreated individuals would strengthen the robustness of the findings.
  1. Statistical Modeling and Performance
  • Dichotomizing the TyG index based on the median may lead to information loss. Analyzing it as a continuous variable or using quantile-based approaches would provide more nuanced insights.
  • Given the inclusion of metabolically related variables, multicollinearity may be present. Reporting Variance Inflation Factor (VIF) values would improve methodological transparency.
  • Since logistic regression was used to predict elevated TyG index, ROC curve analysis and AUC values should be reported to assess the model’s discriminative ability.
  1. Interpretation of Results
  • While the statistical significance of the findings is well presented, the clinical relevance should be discussed. For example, is a 0.3 increase in the TyG index clinically meaningful in terms of risk stratification or intervention?
  1. International Context and Literature
  • The discussion would benefit from comparing the findings with similar studies conducted in other populations (e.g., Latin American, European, East Asian cohorts). This would help contextualize the Mongolian data and assess the generalizability of the results.
  • To ensure the manuscript reflects the current state of research, the authors should consider citing relevant studies from 2024 and 2025, particularly those addressing the prognostic role of the TyG index and its genetic determinants.

Overall Assessment While the association between the TyG index and blood pressure has been previously reported in East Asian populations, this study adds novelty by analyzing a large, nationally representative Mongolian cohort. The finding that even modest elevations in blood pressure are associated with higher TyG index values — particularly when stratified by central obesity — provides new insights into early metabolic risk in this understudied population.

Author Response

(The authors gave the same response as above.)

Reviewer 3 Report

Comments and Suggestions for Authors

Major Concerns

  1. Overinterpretation of Associations
    The authors repeatedly suggest that the TyG index “rises progressively with blood pressure stages” and that “insulin resistance may begin at modest BP increases.” However, the analyses presented are primarily between-group comparisons (normotensive vs. hypertensive categories) rather than true correlation analyses. This risks overstating the strength and nature of the observed relationships.

  2. Categorization of Blood Pressure
    Blood pressure was divided into AHA-defined categories. While clinically useful, this approach reduces statistical power and may create artificial thresholds. Continuous analysis (e.g., regression with systolic and diastolic BP as continuous variables) would provide a more accurate and nuanced understanding of the relationship between BP and TyG.

  3. Confounding by Group Differences
    Hypertensive participants differed substantially from normotensives in age, sex distribution, obesity, diabetes prevalence, and lifestyle factors. Even with adjustment, residual confounding is highly likely. The observed differences in TyG may therefore reflect these comorbidities rather than blood pressure per se.

  4. Lack of Raw Data Presentation
    Presenting raw continuous distributions of BP against TyG (scatterplots, correlation coefficients, regression slopes) would strengthen the argument far more than the current categorical summaries. At present, the analysis does not convincingly establish a true “progressive” association across the full BP spectrum.

Minor Concerns

  • The clinical relevance of the observed TyG differences (often <0.2 units between groups) should be discussed more critically, given the large sample size and potential for statistically significant but trivial effects.

  • Discussion tends to imply causality, which is not supported by the cross-sectional design.

Recommendation
While the dataset is impressive and the research question important, the manuscript in its current form overinterprets group differences as evidence of a graded, independent association. I recommend major revisions, with emphasis on:

  • presenting analyses with BP as a continuous variable,

  • reporting correlation coefficients or regression slopes, and

  • tempering the causal language in the conclusions.

Author Response

(The authors gave the same response as above.)

Round 2

Reviewer 1 Report

Comments and Suggestions for Authors

We acknowledge the effort made in revising the manuscript. The responses provided to the reviewers’ comments were adequate, and the adjustments significantly improved the quality of the article. The current version appropriately addresses the concerns raised and is much clearer and more complete.

Reviewer 2 Report

Comments and Suggestions for Authors

The authors have provided detailed and constructive responses to the raised concerns, and have implemented meaningful revisions to the manuscript. I consider their replies acceptable and recommend proceeding with the editorial process.